# Liquid Chromatography Coupled to High-Resolution Mass Spectrometry for the Confirmation of Caribbean Ciguatoxin-1 as the Main Toxin Responsible for Ciguatera Poisoning Caused by Fish from European Atlantic Coasts

**DOI:** 10.3390/toxins12040267

**Published:** 2020-04-21

**Authors:** Pablo Estevez, Manoella Sibat, José Manuel Leão-Martins, Pedro Reis Costa, Ana Gago-Martínez, Philipp Hess

**Affiliations:** 1Biomedical Research Center (CINBIO), Department of Analytical and Food Chemistry, University of Vigo, Campus Universitario de Vigo, 36310 Vigo, Spain; paestevez@uvigo.es (P.E.); leao@uvigo.es (J.M.L.-M.); 2Ifremer, DYNECO, Laboratoire Phycotoxines, Rue de l’Île d’Yeu, 44311 Nantes, France; manoella.sibat@ifremer.fr; 3IPMA—Portuguese Institute of the Sea and Atmosphere, Av. Brasília, 1449-006 Lisbon, Portugal; prcosta@ipma.pt

**Keywords:** ciguatoxins, HRMS, Q-TOF, ciguatera poisoning, C-CTX1, fragmentation pathways

## Abstract

Ciguatera poisoning (CP) is a common seafood intoxication mainly caused by the consumption of fish contaminated by ciguatoxins. Recent studies showed that Caribbean ciguatoxin-1 (C-CTX1) is the main toxin causing CP through fish caught in the Northeast Atlantic, e.g., Canary Islands (Spain) and Madeira (Portugal). The use of liquid chromatography coupled to tandem mass spectrometry (LC-MS/MS) combined with neuroblastoma cell assay (N2a) allowed the initial confirmation of the presence of C-CTX1 in contaminated fish samples from the abovementioned areas, nevertheless the lack of commercially available reference materials for these particular ciguatoxin (CTX) analogues has been a major limitation to progress research. The EuroCigua project allowed the preparation of C-CTX1 laboratory reference material (LRM) from fish species (*Seriola fasciata*) from the Madeira archipelago (Portugal). This reference material was used to implement a liquid chromatography coupled to high-resolution mass spectrometry (LC-HRMS) for the detection of C-CTX1, acquisition of full-scan as well as collision-induced mass spectra of this particular analogue. Fragmentation pathways were proposed based on fragments obtained. The optimized LC-HRMS method was then applied to confirm C-CTX1 in fish (*Bodianus scrofa*) caught in the Selvagens Islands (Portugal).

## 1. Introduction

Ciguatera poisoning (CP) is a food intoxication mainly related to the consumption of fish contaminated with ciguatoxins (CTXs) [1]. Ciguatoxins (CTXs) are lipophilic cyclic polyethers, which are stable to temperature and accumulate in seafood. CP is endemic in temperate waters of the Pacific Ocean and Caribbean Sea and was recently detected in the Northeast Atlantic [2,3,4,5,6]. A main difficulty in advancing CP research is the lack of reference materials of these toxins and the numerous toxic analogues present in the toxic samples [7,8]. This lack of reference materials limits progress in method development and identification of CTX analogues responsible for the CP. Liquid chromatography coupled to tandem mass spectrometry (LC-MS/MS) is considered the most adequate approach for CTX detection, not only due to the ability for the separation of the different analogues involved in the contamination but also for their sensitive detection by monitoring their mass-to-charge ratio. Several LC-MS/MS methods have been developed for CTX analysis [4,9,10,11]. Research on Caribbean CTXs and in particular on Caribbean ciguatoxin-1 (C-CTX1) as the main toxin responsible for CP in the Caribbean Sea and the Northeast Atlantic has been limited [12,13]. Despite this limitation, efforts have been recently made to implement LC-MS/MS methods for the detection of CTXs, particularly in the areas of concern [14,15]. Alternative approaches such as the combination of neuroblastoma cell assay (N2a) with LC-MS/MS have also been used as a powerful tool for the confirmation of C-CTX1 and different C-CTX congeners [16]. 

The potential for confirmation provided by high-resolution mass spectrometry (HRMS) makes this approach necessary, in particular when the availability of reference materials is limited. Nevertheless, the lack of sensitivity is a challenge that compromises its applicability as concentrations of CTXs are typically low. The complexity of the matrix and the difficulty in the ionization of CTXs also jeopardizes the applicability of LC-HRMS for the detection of CTXs. Despite this, the applicability of HRMS has been demonstrated in particular for CTX1B and CTX3C as the main toxins responsible of CP in the Pacific Ocean, the fragmentation pathways have been identified, increasing the knowledge on how these complex toxins fragment in the mass spectrometer [17,18,19]. Recent studies published by Kryuchkov et al. [20] reported strategies for the structure elucidation of Caribbean ciguatoxin analogues (C-CTX3 and C-CTX4). 

This study responds to one of the objectives included in the EUROCIGUA project, which focused on the risk characterization of ciguatera food poisoning in Europe, and it is partially funded by the European Food Safety Authority (EFSA). One of the objectives of the project is the confirmation by HRMS of the CTX analogues previously detected by LC-MS/MS in fish samples from selected areas in the EU. To accomplish this objective, an LC-HRMS method was developed to obtain C-CTX1 fragmentation pathways, which will allow the characterization of the C-CTX1 molecule. The method was subsequently applied to confirm the presence of C-CTX1 in a contaminated fish sample from the Northeast Atlantic (Selvagens Islands, Madeira archipelago, Portugal), previously identified by LC-MS/MS.

## 2. Results and Discussion

### 2.1. LC-HRMS Analysis in Full-Scan MS Mode of C-CTX1

A laboratory reference material (LRM) of C-CTX1 isolated from amberjack, *Seriola fasciata*, was analyzed by LC-HRMS. The use of aqueous methanol as mobile phase was dismissed due to the formation of highly stable [M + Na]^+^ and the difficulty in its fragmentation. For this reason, the LC and ion source conditions proposed by [15] for C-CTX1 confirmation monitoring water losses and specific fragments in the presence of aqueous acetonitrile were followed.

LC-HRMS in full-scan MS acquisition mode detected C-CTX1 at a retention time of 7.8 min with an ion pattern where the predominant ion was the first water loss *m/z* 1123.6184 (−4.5 ppm) [M + H - H_2_O]^+^ followed by the sodium adduct *m/z* 1163.6106 (−1.6 ppm) [M + Na]^+^. The protonated molecule [M + H]^+^ as well as ammonium [M + NH_4_]^+^ and potassium adducts [M + K]^+^ were present but with a lower intensity and with higher mass differences (>10 ppm) (Figure 1A). The assigned positive HRMS ion species and their corresponding mass errors (ppm) are presented in Figure 1A.

### 2.2. LC-HRMS Analysis in Targeted MS/MS Mode of C-CTX1

C-CTX1 fragmentation pathways were proposed based on HRMS/MS spectra in positive targeted MS/MS mode with a range of six collision energies (from 20 to 70 eV). In HRMS spectra (Figure 1A), the protonated ion [M + H]^+^ was not the predominant ion cluster; so due to its high intensity, the first water loss *m/z* 1123.6200 [M + H - H_2_O]^+^ was selected as precursor ion to be fragmented. 

MS/MS spectra of C-CTX1 selecting *m/z* 1123.6200 [M+H-H_2_O]^+^ were characterized by the detection of successive water losses at *m/z* 1105.6094 [M+H−2H_2_O]^+^, *m/z* 1087.5989 [M + H - 3H_2_O]^+^, and *m/z* 1069.5883 [M + H - 4H_2_O]^+^ (Figure 1B and Appendix A). As with CTX3C and CTX1B [17], the opening of the G- and H- rings allowed the formation of the ion at *m/z* 563.3220 followed by water losses at *m/z* 545.3115, *m/z* 527.3009, and *m/z* 509.2903, these fragments were also detected in [20] (Figure 1B and Appendix A). Fragmentation in K-, L-, M-, and N-rings gave rise to several fragments with water losses at *m/z* 253.1439 and *m/z* 235.1333 as reported in [20] and *m/z* 181.1229 as well as the fragment *m/z* 147.1021 followed by the loss of a –CH_2_ group *m/z* 133.0865. A-, B-, C-, and D- rings were also prone to fragmentation showing ions with water losses and the loss of a –CH_2_ group at *m/z* 209.1178, *m/z* 191.1072, *m/z* 153.0916, *m/z* 103.0759, *m/z* 89.0603, and *m/z* 59.0497 (Figure 1B and Appendix A). LC-HRMS allowed to confirm the exact mass as well as the fragmentation pathway of C-CTX1 fragments previously detected in LC-LRMS [15,21].

These results confirmed that the C-CTX1 molecular ion with theoretical *m/z* 1141.6306 [M + H]^+^ tends to be fragmented in the source compromising its detection. Therefore, the first water loss *m/z* 1123.6200 [M + H - H_2_O]^+^ is the highest intense ion formed in C-CTX1 ionization when using acetonitrile with aqueous ammonium formate and formic acid as mobile phase. This is in agreement with previous studies where C-CTX1 first water loss is selected as precursor ion in SRM (selected reaction monitoring) or MRM (multiple reaction monitoring) mode [4,15]. 

### 2.3. Analysis of a Naturally Contaminated Sample from Selvagens Islands (Madeira Archipelago, Portugal)

Barred hogfish (*Bodianus scrofa*) from Selvagens Islands (Madeira, Portugal) was extracted and purified following the conditions described by [15]. The presence of C-CTX1 was confirmed by LC-HRMS under the conditions described in this work. C-CTX1 was detected at the same retention time as the C-CTX1 LRM (7.8 min) (Appendix A). Ion pattern confirmed the detection of C-CTX1 with the first water loss *m/z* 1123.6214 (+1.2 ppm) [M + H - H_2_O]^+^ and the sodium adduct at *m/z* 1163.6138 (+1.1 ppm) [M + Na]^+^ (Appendix A). Despite the detection of C-CTX1 by LC-HRMS and the good mass accuracy obtained in full-scan MS mode, sensitivity was not enough to obtain MS/MS spectra.

## 3. Conclusions

The results obtained in this study show the potential of LC-HRMS for the confirmation of the presence of C-CTX1 in fish from the European Atlantic coasts. C-CTX1 full-scan MS and MS/MS spectra were successfully obtained and fragmentation pathways showed a similar fragmentation pattern compared to CTX3C and CTX1B, with fragmentation, not only in the ends of the molecule but also in the G- and H- rings. The confirmation of C-CTX1 by LC-HRMS is in agreement with previous studies carried out by combination of N2a and LC-MS/MS in which C-CTX1 had been confirmed as the main compound responsible for the CTX contamination in EU Atlantic Coasts. The relevance of the results obtained in this study are justified not only by the accomplishment of an important objective of the EUROCIGUA project in which the authors of this work are involved, but also for having contributed to the characterization of CP in areas where this contamination is being considered an emerging issue. 

## 4. Materials and Methods

### 4.1. Reference Materials and Samples

C-CTX1 LRM (estimated to 123 ng/mL) purified from amberjack *S. fasciata* from Madeira archipelago (Portugal) was prepared at the University of Vigo, in the framework of the Specific Grant 4 of the project EuroCigua (the preparation of these materials is fully described on a manuscript in preparation, by Gago Martínez and coauthors). C-CTX1 in the LRM was identified by comparison with authentic C-CTX1 kindly provided by Robert W. Dickey (previously, U.S. Food and Drug Administration) via Ronald Manger (Fred Hutchinson Cancer Research Center, Seattle, WA, USA).

Barred hogfish (*B. scrofa)* was captured in the marine protected area of Selvagens Islands (Madeira, Portugal) under the EUROCIGUA framework agreement in September 2018. A portion of flesh tissue was dissected and stored raw at −20 °C until use. 

### 4.2. Reagents

Acetone, water, methanol, hexane, diethyl ether, and ethyl acetate used for sample preparation were HPLC grade (Merck KGaA, Darmstadt, Germany).

Water, acetonitrile, formic acid, and ammonium formate used to prepare mobile phases were of LC-MS grade. All these chemicals were purchased from Sigma Aldrich (Saint Quentin Fallavier, France).

### 4.3. Sample Pretreatment

Barred hogfish was extracted and purified following the conditions proposed by [15]. 

### 4.4. LC-HRMS and HRMS/MS (Q-TOF 6550 iFunnel)

LC-HRMS analyses were carried out using a UHPLC system 1290 Infinity II (Agilent Technologies, Santa Clara, CA, USA) coupled to a HRMS quadrupole/time-of-flight (Q-TOF) mass spectrometer, i.e., Q-TOF 6550 iFunnel (Agilent Technologies, Santa Clara, CA, USA). Chromatographic separation was performed using a Kinetex C18 column (100 Å, 1.7 µm, 100 × 2.1 mm, Phenomenex, Le Pecq, France) at 40 °C with 5 mM ammonium formate and 0.1% formic acid in water (A) and acetonitrile (B). The flow rate was set at 0.4 mL/min and the injection volume was 5 µL. Gradient of mobile phase was carried out as follows: 35% B was kept for 1 min increasing to 80% B in 15 min, 95% of B in 16 min keeping for 5 min, and returning to the initial conditions in 0.1 min equilibrating the column for 4.9 min prior to the next injection.

LC-HRMS analyses were carried out in full-scan and targeted MS/MS mode in positive ionization. Full-scan analysis operated at a mass resolution of 40,000 full width at half maximum (FWHM) over a mass-to-charge ratio (*m/z*) ranging from 800 to 1200 with a scan rate of 1 spectrum/s. Targeted MS/MS was performed in a collision induced dissociation (CID) cell at 45,000 FWHM over the scan rage from *m/z* 50 to 1200 with a scan rate of 10 spectra/s and a scan rate of 3 spectra/s applying three different collision energies in order to have a good fragmentation pathway. A reference mass of *m/z* 922.0099 (hexakisphosphazine) was continuously monitored during the entire run. Acquisition was controlled by MassHunter software (Agilent technologies, CA, USA). Raw data were processed with Agilent MassHunter Qualitative Analysis software (version B.07.00, service pack 1) using the Find by Formula (FbF) algorithm screening with a Personal Compound Database and Library (PDCL) created by the Phycotoxins Laboratory (IFREMER, Nantes, France).

Source conditions were as follows: gas temperature, 250 °C; gas flow, 16 L/min; nebulizer, 15 psi; sheath gas temperature, 400 °C; sheath gas flow, 12 L/min; capillary voltage, 5000 V, and nozzle voltage, 1000 V. The instrument was calibrated, using the Agilent tuning mix, in positive ionization mode before each analysis. 

## Figures and Tables

**Figure 1 toxins-12-00267-f001:**
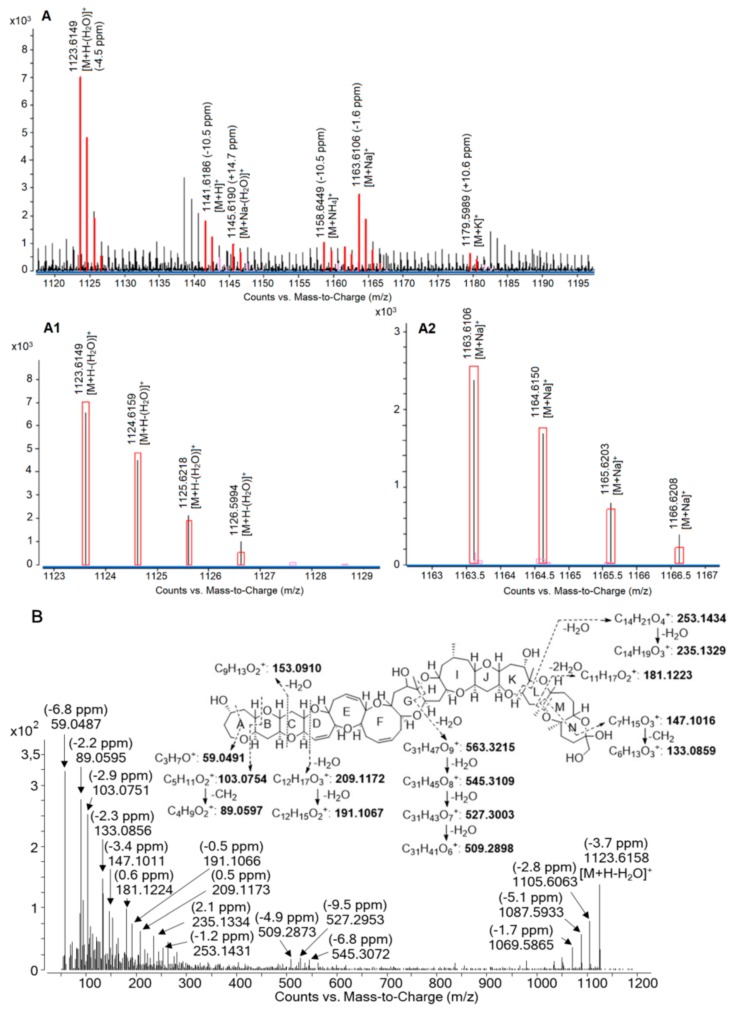
(**A**) Positive electrospray ionization (ESI^+^) full-scan MS spectra of Caribbean ciguatoxin-1 (C-CTX1) laboratory reference material (LRM) (estimated to 123 ng/mL); (**A1**) zoom of C-CTX1 *m/z* 1123.6149 [M + H - H_2_O]^+^; (**A2**) zoom of C-CTX1 *m/z* 1163.6106 [M + Na]^+^ (rectangles in red represent the theoretical isotope abundance and spacing of C-CTX1). (**B**) ESI^+^ targeted MS/MS spectra of C-CTX1 LRM (estimated to 123 ng/mL) selecting C-CTX1 *m/z* 1123.6200 [M + H - H_2_O]^+^ at an average collision energy of 30, 50, and 70 eV.

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
