# Peer review of "Liquid Chromatography Coupled to High-Resolution Mass Spectrometry for the Confirmation of Caribbean Ciguatoxin-1 as the Main Toxin Responsible for Ciguatera Poisoning Caused by Fish from European Atlantic Coasts"

_toxins, 2020, doi:10.3390/toxins12040267_

Round 1
Reviewer 1 Report
Liquid Chromatography coupled to High Resolution Mass Spectrometry for the confirmation of Caribbean Ciguatoxin-1 as the main toxin responsible for Ciguatera Poisoning in fish from European Atlantic coasts
The manuscript describes the use of High Resolution mass spectrometry for the confirmation of Caribbean ciguatoxin-1, a potent toxin in fish contaminated with Caribbean ciguatoxins. Analytical standard of the toxin is not commercially available and thus confirmation of the toxin in fish is challenging. This work provides a method to confirm C-CTX-1 by HRMS and provides value in ciguatera research. Manuscript is well organized and well presented.
Few edits recommended are:
Line 5: (Title) …Ciguatera poisoning in fish? May need to edit as “….responsible for Ciguatera Poisoning caused by fish from ..”
Line 7: Is CP an accepted acronym for Ciguatera Poisoning?
Line 15: edit : …preparation of C-CTX-1 laboratory reference material (LRM) from fish…..
Line 30: edit: “….. and can accumulate in fish” (comments: It is not accumulated just in flesh. Also, since the authors took “Fish” out from Ciguatera Fish Poisoning, it would be reasonable to generalize that accumulation is in seafood rather than in fish.)
Line 55-57: This sentence is not very clear. “One of the objectives of the project is the confirmation by HRMS of the CTXs involved in the CP contamination in the EU selected areas which had been previously detected by LC-MS/MS”. Are the samples from CP incidents?
May need to edit as: “One of the objectives of the project is the confirmation by HRMS the CTXs previously detected by LC-MS/MS in samples from CP incidents in selected areas in the EU”
Line 78-79: “….protonated ion [M+H]+ was not the predominant ion cluster; so due to its high intensity, the first water loss m/z 1123.6200 [M+H-H2O]+ was…..
Page 3, Fig 1: A) m/z range 1175-1185 is covered with the label A2;
Wrong label B2 on A2 spectrum
Line 99: “…highest intense ion formed ….
Line 101-102: “ …. previous studies where C-CTX1 first water loss is selected as precursor ion in SRM (Single Reaction Monitoring) or MRM….”
Please edit the sentence. SRM (Selected Reaction Monitoring)
Line 107: “…. Ion pattern confirmed the detection of C-CTX-1 with the first water loss m/z 1123.6214 (+1.2 ppm) [M+H-H2O]+ and the ……
Line 109: “….. the detection of C-CTX1 by LC-HRMS and the good mass accuracy obtained in full scan MS …..”
Line 119: “ … accomplishment of an important objective of the EUROCIGUA project in which the authors of this work are involved, but also for having ….
Line 145: “ … The flow rate was set at …..
Author Response
The manuscript describes the use of High Resolution mass spectrometry for the confirmation of Caribbean ciguatoxin-1, a potent toxin in fish contaminated with Caribbean ciguatoxins. Analytical standard of the toxin is not commercially available and thus confirmation of the toxin in fish is challenging. This work provides a method to confirm C-CTX-1 by HRMS and provides value in ciguatera research. Manuscript is well organized and well presented.
The authors thank the reviewer for the comments and reply to specific comments below.
Few edits recommended are:
Line 5: (Title) …Ciguatera poisoning in fish? May need to edit as “….responsible for Ciguatera Poisoning caused by fish from ..”
The title was modified according to the edit suggested by the reviewer.
Line 7: Is CP an accepted acronym for Ciguatera Poisoning?
The term Ciguatera Poisoning and its acronym CP is the term currently accepted for the scientific community, as the general term in order to include not only poisoning caused by fish but also the one caused by other marine species susceptible to be contaminated with ciguatoxins.
Line 15: edit : …preparation of C-CTX-1 laboratory reference material (LRM) from fish…..
The sentence was modified.
Line 30: edit: “….. and can accumulate in fish” (comments: It is not accumulated just in flesh. Also, since the authors took “Fish” out from Ciguatera Fish Poisoning, it would be reasonable to generalize that accumulation is in seafood rather than in fish.)
We agree with the reviewer and “fish” has been modified by “seafood”.
Line 55-57: This sentence is not very clear. “One of the objectives of the project is the confirmation by HRMS of the CTXs involved in the CP contamination in the EU selected areas which had been previously detected by LC-MS/MS”. Are the samples from CP incidents?
The samples analyzed during the EUROCIGUA project were mostly samples collected according to sampling plans established in the project in specific areas of the EU where the incidence of CP was suspected. Only a few samples correspond to CP incidents. After a previous screening of toxicity by N2a cell assay, the first confirmation of the CTX toxicity was carried out by LC-MS/MS which allowed to identify the main CTX analogues responsible for the contamination. A further confirmation was carried out by LC-HRMS in order to confirm the identity of the CTX s analogues previously detected by LC-MS/MS.
May need to edit as: “One of the objectives of the project is the confirmation by HRMS the CTXs previously detected by LC-MS/MS in samples from CP incidents in selected areas in the EU”
Since the majority of the samples analyzed correspond to specific sampling plans and only a few ones correspond to CP incidents the sentence was modified to: “One of the objectives of the project is the confirmation by LC-HRMS of the CTX analogues previously detected by LC-MS/MS in fish samples from selected areas in the EU.”
Line 78-79: “….protonated ion [M+H]+ was not the predominant ion cluster; so due to its high intensity, the first water loss m/z 1123.6200 [M+H-H2O]+ was…..
Page 3, Fig 1: A) m/z range 1175-1185 is covered with the label A2;
Wrong label B2 on A2 spectrum
Line 99: “…highest intense ion formed ….
Line 101-102: “ …. previous studies where C-CTX1 first water loss is selected as precursor ion in SRM (Single Reaction Monitoring) or MRM….”
Please edit the sentence. SRM (Selected Reaction Monitoring)
Line 107: “…. Ion pattern confirmed the detection of C-CTX-1 with the first water loss m/z 1123.6214 (+1.2 ppm) [M+H-H2O]+ and the ……
Line 109: “….. the detection of C-CTX1 by LC-HRMS and the good mass accuracy obtained in full scan MS …..”
Line 119: “ … accomplishment of an important objective of the EUROCIGUA project in which the authors of this work are involved, but also for having ….
Line 145: “ … The flow rate was set at …..
All the modifications suggested by the reviewer were taken into consideration

Reviewer 2 Report
The manuscript described fragmentation pathways of C-CTX1 on LC-HRMS analysis. This finding will be a useful tool for structure elucidations of other C-CTX analogues suspected to be present in the toxic fish specimens.
Recently, it was published the manuscript entitled “LC–HRMS and Chemical Derivatization Strategies for the Structure Elucidation of Caribbean Ciguatoxins: Identification of C-CTX-3 and -4” by Kryuchkov et al. in Marine Drugs 2020, 18, 182 (doi:10.3390/md18040182). Although, both manuscripts seem to be similar, this manuscript described more in detail and new finding of the fragmentation pathways related to ring A to D. This finding will contribute to elucidate the structure of ring A-D derivatives in future studies. It is recommended to cite Kryuchkov et al. (2020) and revise the manuscript with this reference.
Other comments are as followings:
L64
“reference material” --> “laboratory reference material (LRM)”?
L65
“aqueous methanol” --> “aqueous methanol as mobile phase”?
Figure 1
Consider the figures divide into two (A as 1 and B as 2) and expand the structure and fragmentation passway of C-CTX1 to be conspicuous.
Figure legend of B is missed.
“A2” should be on the “B1”
No peak was present for m/z 1164.6150 on A2.
Explain what does red edged bar mean.
L110
Show the chromatogram of LRM and contaminated sample. In case there is no space due to limitation of the pages, it’ll be fine to be as supplementary material.
L125-127
Explain how C-CTX1 in LRM was identified, LC-MS? It is difficult to judge the reliability of the LRM under the review, since authors names are blinded.
I think there are only three groups those have authentic C-CTX1, Bob Dicky and Abraham Ann from US, and Richard Lewis from Australia. If the C-CTX1 was identified by comparison with authentic C-CTX1, please explain the origin of the authentic C-CTX1, if the researcher mentioned above was not included as co-author.
L145
“ret” --> “rate”?
Author Response
The manuscript described fragmentation pathways of C-CTX1 on LC-HRMS analysis. This finding will be a useful tool for structure elucidations of other C-CTX analogues suspected to be present in the toxic fish specimens.
The authors thank the reviewer for the comments: Please find below the replies to the specific comments
Recently, it was published the manuscript entitled “LC–HRMS and Chemical Derivatization Strategies for the Structure Elucidation of Caribbean Ciguatoxins: Identification of C-CTX-3 and -4” by Kryuchkov et al. in Marine Drugs 2020, 18, 182 (doi:10.3390/md18040182). Although, both manuscripts seem to be similar, this manuscript described more in detail and new finding of the fragmentation pathways related to ring A to D. This finding will contribute to elucidate the structure of ring A-D derivatives in future studies. It is recommended to cite Kryuchkov et al. (2020) and revise the manuscript with this reference.
The following sentence was included in the introduction to cite the reference suggested by the reviewer. Line 51-52: « Recent studies published by Kryuchkov et al. [20] reported strategies for the structure elucidation of Caribbean Ciguatoxins analogues (C-CTX3 and C-CTX4). »
Line 81-86: “ ….the opening of the G- and H- rings allowed the formation of the ion at m/z 563.3220 followed by water losses at m/z 545.3115, m/z 527.3009 and m/z 509.2903, these fragments were also detected in [20] (Figure 1B & S1, Table S1). Fragmentation in K-, L-, M- and N-rings gave rise to several fragments with water losses at m/z 253.1439, m/z 235.1333 as reported in [20]….”
Other comments are as followings:
L64
“reference material” --> “laboratory reference material (LRM)”?
Modified.
L65
“aqueous methanol” --> “aqueous methanol as mobile phase”?
Modified.
Figure 1
Consider the figures divide into two (A as 1 and B as 2) and expand the structure and fragmentation passway of C-CTX1 to be conspicuous.
We agree with the reviewer, but the number of figures in Toxins Communications is limited and only one figure is allowed
Figure legend of B is missed.
Included.
“A2” should be on the “B1”
Modified.
No peak was present for m/z 1164.6150 on A2.
A peak is detected for m/z 1164.6150 on A2, maybe there is some problem with the Word processor of the reviewer?
Explain what does red edged bar mean.
The explanation of the red edged bars was included in the legend of figure 1. Lines 94-95: “(rectangles in red represent the theoretical isotope abundance and spacing of C-CTX1) “
L110
Show the chromatogram of LRM and contaminated sample. In case there is no space due to limitation of the pages, it’ll be fine to be as supplementary material.
The chromatograms were included in the supplementary material.
L125-127
Explain how C-CTX1 in LRM was identified, LC-MS? It is difficult to judge the reliability of the LRM under the review, since authors names are blinded.
I think there are only three groups those have authentic C-CTX1, Bob Dicky and Abraham Ann from US, and Richard Lewis from Australia. If the C-CTX1 was identified by comparison with authentic C-CTX1, please explain the origin of the authentic C-CTX1, if the researcher mentioned above was not included as co-author.
The referee is right and the standard used in this work was kindly provided by Robert Dickey. In order to make clear the source of the standard the text was modified and the following sentence was included.
The following sentence was included in the text. Line 129-131: “ C-CTX1 in the LRM was identified by comparison with a pure standard of C-CTX1 kindly provided by Robert W. Dickey (previously, U.S. Food and Drug Administration) via Ronald Manger (Fred Hutchinson Cancer Research Center, Seattle, USA).”
This sentence is also included in the Acknowledgements section which is not visible in the version sent to reviewers.
L145
“ret” --> “rate”?
Modified.